# Still’s Disease in the Constellation of Hyperinflammatory Syndromes: A Link with Kawasaki Disease?

**DOI:** 10.3390/jcm10153244

**Published:** 2021-07-23

**Authors:** Perrine Dusser, Isabelle Koné-Paut

**Affiliations:** 1Paediatric Rheumatology Department, Université Paris-Saclay, APHP, Bicêtre Hospital, 94270 Le Kremlin-Bicêtre, France; perrine.dusser@aphp.fr; 2Centre de Référence des Maladies Auto-Inflammatoires et des Amyloses Inflammatoire (CEREMAIA), Université Paris-Saclay, APHP, Bicêtre Hospital, 94270 Le Kremlin-Bicêtre, France

**Keywords:** Kawasaki disease, Still’s disease, disease overlap, interleukin-1

## Abstract

Still’s disease and Kawasaki disease (KD) today belong to the group of cytokine storm syndromes, a pathophysiological set related to excessive activation of the innate immune response. We present here a personal vision of what can link these two diseases, taking up their concepts at their beginning. By their many clinical and physiopathological similarities, we conclude that they constitute a common spectrum whose fate is modified by subtle differences in terms of adaptive response that could, in part, be driven by genetic factors.

## 1. Conceptual Evolution of Still’s Disease

Sir Frederick Still described at the end of the 19th century (1897), a primary rheumatic condition remarkably different from rheumatoid arthritis with “a chronic progressive enlargement of joints, associated with general enlargement of glands and enlargement of spleen” and with the onset “in young children almost always before the second dentition.” He also noted in his report the presence of fever, chills, anaemia and stunted growth in a subset of patients [1]. It was not until 1946, with the advent of paediatric rheumatology by pioneers such as Mayer S. Diamantberger and Barbara Ansell, that the heterogeneity of juvenile arthritis was described, in which Still’s disease appeared as a separate subgroup [2,3]. The major distinguishing features of Still’s disease, in addition to those previously described, were the severity of joint involvement in 50% of cases with predominant involvement of the hips, knees and wrists. It was during the 1980s that a first alert on cases of severe hepatic and neurological complications, occurring after an infection or the use of certain drugs in children, made it possible to make the association between Still’s disease and macrophage activation syndrome (MAS), a peculiar entity neighbour of inherited hemophagocytic lymphohistiocytosis (HLH) [4]. Over time, a very specific treatment response profile was also identified, with frequent dependence on corticosteroids, little benefit from treatment with methotrexate and later with anti TNF, the first biotherapies available for JIA in the early 1990s [5]. This resistance to treatments echoed the work of Dayer and de Benedetti who had pointed out years earlier the predominant role of interleukin 6 (IL-6) in the disease pathogenesis [6,7]. Other studies then highlighted the role of interleukin 1 (IL-1) in systemic manifestations, and first attempt of IL-1 blockade by anakinra have triggered hopes and expectations [8].

In parallel with all this, the concept of auto inflammation emerged in 1999 with the discovery of the TNF receptor 1 (TNFR1) deficiency syndrome, an inherited hyper inflammatory condition, linked to aberrant and deregulated IL-1 secretion induced by intracellular stress related to a defect in folding and clearance of the mutated TNFR1 [9]. Thus, other hereditary and multifactorial diseases, presenting with similar inflammatory characteristics, mediated by IL-1, gradually entered this category. Briefly, patients experience acute or chronic inflammatory symptoms including fever, malaise, myalgia, skin rashes, and abdominal pain. The biological inflammatory syndrome appears pure (without autoimmunity) and is very marked during relapses (polynucleosis, increased CRP and serum amyloid A: SAA). A number of these auto-inflammatory diseases share common underlying mechanism in which, mutations affect intracellular receptors of innate immunity, involved in the assembly of inflammasome complexes (multiprotein complexes), that leads to activation of the caspase 1, and enhance an excessive and deregulated secretion of IL-1 and IL-18 [10]. Interleukin-1 blockade has become the gold standard treatment for this group of diseases. It did not take very long to see the phenotypic similarity between the inflammasomopathies group and Still’s disease; these diseases having also in common, the young age of onset and the possible, although rare, progression to MAS and AA amyloidosis. With time and the dramatic development of new molecular genetic techniques, the field of systemic autoinflammatory diseases (SAID) has expanded to include other mechanisms related to, for example, type I interferon and NFKB pathways. However, in all cases, the hyper inflammatory phenotype, even when associated with a note of immune deficiency or autoimmunity, remains in the foreground.

## 2. Conceptual Evolution of Kawasaki Disease

In 1961 in Japan, in a 4-year-old boy with a 2-week fever, Dr Tomisaku Kawasaki saw for the first time non-purulent conjunctivitis, red, dry, and fissured lips, a raspberry tongue, cervical lymphadenopathy, erythema multiforme, and changes in the extremities (redness, swelling of the palms and soles). It was not until another 50 cases were described that a new disease was recognized and named after him [11]. The associated coronary artery aneurysms (CAA) were not identified until 1970. Initially, Kawasaki disease (KD) appeared to be an infectious disease, but to date, no single causative agent has been identified. Before intravenous immunoglobulins (IVIG) treatment became the gold standard, the incidence of CAA was 25% [12].

KD is a systemic inflammatory disease affecting primarily the medium-sized vessels and occurring in more than 75% of cases before the age of 5 years. The disease generally follows or is concomitant of upper respiratory or gastrointestinal symptoms, and manifests as persistent spiking fevers; associated with muco cutaneous symptoms (generalized skin rash, eye redness, strawberry tongue, oral cavity erythema, redness and edema at the hands and feet, and enlarged cervical lymph nodes) [13]. At onset, a biological cytokine storm containing, IL-1, IL-6, TNF, IL-17A, chemokines and metalloproteases, is sometimes present going as far as MAS, and its intensity goes hand in hand with the development of CAA, the most feared complication [14]. Studies carried out on the murine model of KD vasculitis have led to a better understanding of the role of IL-1 and the NLRP3 inflammasome in the induction of endothelial cells injury causing aneurysms [15]. The role of IL-1 in human KD is now largely supported and the IL-1 blockade could be one of the mechanisms of action of IVIG, the standard treatment of KD. In addition, exploratory trials are developing for better defining the full effects and the place of IL-1 blocking agents in KD treatment strategy [16].

## 3. Still’s Disease and KD, How Much Are They Related?

### Is It Possible to Link Them Clinically?

Although it is not clear what triggers them, it appears that Still’s disease and KD occur in the same age groups with a maximum of cases around the age of 2 to 4 years and rarer cases at adolescence and in young adults [17,18,19,20]. Both correspond to a deregulation of the innate immune response towards an excessive inflammation following a still unknown stimulation, coming from the airways or the digestive system. The involvement of the oropharyngeal sphere is probably a critical point that could link Still ‘s disease and KD to another multifactorial autoinflammatory syndrome, namely PFAPA syndrome, which also occurs in the same age groups [21]. Other clinical symptoms, spiking fever, rash, hepatomegaly, adenopathy, and cardiac involvement are very similar between KD and Still’s disease (Table 1).

In addition, it is not uncommon in the paediatric experience for patients to develop Still’s disease after an initial presentation of KD including coronary dilations [25]. Patients in both diseases develop severe biological inflammation with hyperleukocytosis, thrombocytosis, anaemia, hypoalbuminemia, and hepatic cytolysis. Other biomarkers of inflammation such as, ferritin, IL-1β, IL-6, TNF, IFNγ and S100 proteins are elevated and correlate with disease activity in both diseases and cardiac complications in KD [26,27] (Table 2). MAS is almost a hallmark of Still’s disease in which, it is present subclinically in 40% of cases and apparent in 7% [17]. MAS may also occur during the acute phase of KD with an estimate of 1% of cases; the actual incidence is still unknown because it is more often subclinical and exceptionally apparent and life threatening [28].

## 4. How Their Pathophysiology Could Be Similar?

### Innate Immune System Activation

Both diseases involve a cytokine storm, a type of aberrant innate immune response as described in systemic forms of COVID-19 (multisystem inflammatory syndrome) and shared by other as yet poorly understood conditions such as SAM, HLH, cytokine release syndrome, toxic shock syndrome and acute respiratory distress syndrome [32]. Infectious organisms are capable of triggering onset of both diseases in genetically predisposed children, affected mainly between 6 months and 5 years of age. The peak age of diseases onset coincides with the period during which children are most susceptible to common pathogens. Microbial danger signals (PAMPs) recognize innate immune receptors (TLRs, NOD2, NLRP3) that enhance numerous danger signals (DAMPs: S100 proteins, metalloproteases, pro IL-1β) and secondary activation of canonic pathways cytokines secretion, in which both the NFKB and inflammasome pathways are involved (Figure 1) [33]. Among genetic susceptibility factors of KD, single-nucleotide polymorphisms in *ITPKC* encoding for an inositol-trisphosphate 3-kinase, regulating negatively the opening of calcium channels, induce an increase of calcium release with two downstream consequences; 1/the activation of the calpain pathway enhancing the interleukin 1α a potent danger signal and 2/the activation of the NLRP3 inflammasome secondary to increased secretion of ROS (reactive oxygen species) [34]. Susceptibility to Still’s disease and response to IL-1 blockage treatment have recently been linked with IL1-receptor antagonist gene (IL1RN) polymorphisms [35]. MAS is a common complication in both diseases and is generally linked with either: onset of disease, disease activity or coinfection as a second hit.

## 5. How Much Are They Different?

### 5.1. Clinical Differences

What really differentiates the two conditions is their spontaneous evolution and their target organs. The evolution of the KD is spontaneously limited in time, exceptionally recurrent and its main target is the medium-sized vessel (to a lesser degree the small vessel). Still’s disease has an unremitted course in 50% of cases and polyphasic in 25%; over time, it mainly progresses to arthritis. Response to treatment may be different and early treatment with IVIG is effective in 80% of KD patients but not in those with Still’s disease. High dose corticosteroids alleviate systemic symptoms in both diseases. The use of early interleukin-1 blockade treatment could be the first choice in both diseases but still needs further investigation, especially in KD.

### 5.2. Adaptive Immune System Induction

While the two diseases appear having a common background of activation of innate immunity, their pathways of secondary induction of adaptive immunity might be different, suggested by their distinct evolutionary courses. In both cases, the activation of the NFKB pathway induces IL-6, IFNγ and IL-17A, among others. The IL-17A plays a major role in perpetuating inflammation in various cellular types such as fibroblasts, macrophages, endothelial cells, and chondrocytes [14].

An important point, which could significantly modify the clinical expression of the adaptive inflammatory phenotype, would be the participation of modifiers in this immune response, among which genetic factors, probably not unique (Figure 2).

As the ITPKC gene product negatively regulates T-cell activation, the presence of an ITPKC variant, as found in patients with KD, might lead to uncontrollable T-cell activation, through NFAT activation, which in turn might lead to elevated IL-17 levels. In addition, the calcium-dependent calpain may induce more IL-1α precursor. Together with IL1-β, the IL-1α plays an important role in the development of KD vasculitis [33]. Another important factor of development of vasculitis is probably the IgA. An increased number of IgA-producing plasma cells in tissues and coronary artery vascular walls have been found in affected patients with KD. In addition, a murine model of KD vasculitis is dependent on intestinal barrier dysfunction leading to secretory IgA leakage and IgA-C3 immune complex deposition in cardiovascular lesions [36]. In paediatric onset Still ‘s disease, low expression of IL1RN and IL-1 receptor antagonists SNPs correlated with increased risk of disease and lower response to anakinra treatment [35].

### 5.3. Endothelial Activation

Endothelial activation is more prominent in KD than in Still’s disease, as demonstrated by the higher frequency of coronary vasculitis and myocardial inflammation in KD. In KD, the involvement of coronary arteries reaches a frequency of 25% of untreated patients and may be up to 50% in the youngest before the age of one year. In addition to vessel wall inflammation, endothelial cells dysfunction and impaired remodelling contribute to aneurysms, thrombosis and secondary stenosis. Several studies have demonstrated the role of an IL-1b and IL-6 inducible cytokine, the vascular endothelial growth factor (VEGF), whose secretion is increased by KD patient’s neutrophils at the acute phase of the disease, and is correlated with the development of CAA [37] In addition, a partly genetically determined unbalance between two angiogenic factors: angiopontin 1 (Ang1) and Ang2 toward Ang1 down regulation, together with the increase of VEGF, promotes endothelial inflammation [38].

Even if this remains hypothetical, the role played by IL-1α in KD could be distinctive. Indeed, IL-1α is a potent modulator of endothelial cell-surface properties and function, and is released after cell necrosis playing the role of an alarmin then inducing early inflammation [39].

## 6. Conclusions

There is a clinical and pathophysiological obvious overlap between Still’s disease and KD. Their cause is probably not unique, but they probably occur following the same types of triggers in individuals carrying various genetic predisposing factors acting in concert.

## 7. Take Home Messages

### 7.1. What Are the Key Elements That Our Analysis Points Out?

–KD and Still’s disease share a number of clinical and biological similarities but essentially differ from their respective evolutions.–Both diseases are underpinned by related physiopathological mechanisms encompassing a potent activation of innate immunity.–Rather, they should be considered as belonging to a common clinical spectrum, the extremes of which depend on a variable influence of adaptive immunity.–The preferential endothelial activation of KD compared to Still’s disease is strongly enhanced by the release of interleukin 1 alpha and the direct toxicity of the IgA toward the endothelial cells.

### 7.2. How Might This Impact on Clinical Practice or Future Developments?

–The deep link between these two syndromes (rather than diseases) implies considering them together to resolve physiopathological or therapeutic research questions.–The crucial role of interleukin 1 alpha in inducing cardiac inflammation of KD must be taken into account when choosing targeted biotherapies.

## Figures and Tables

**Figure 1 jcm-10-03244-f001:**
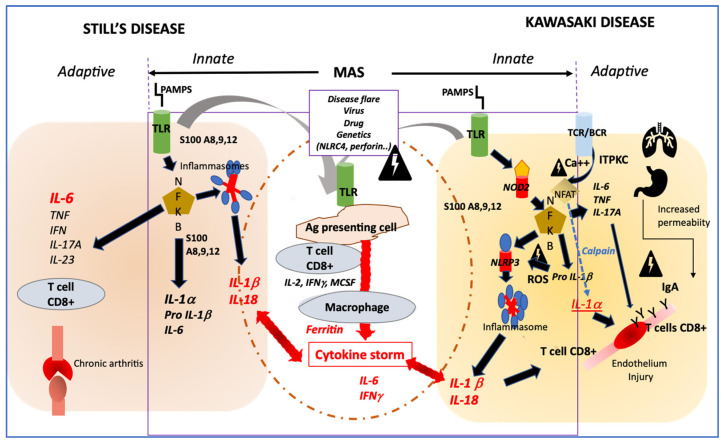
Profiles of immune activation are similar in Still’s disease and Kawasaki disease. *Innate immune system*: In both diseases danger signals coming from outside (PAMPS) sensed by TLR or from inside (DAMPS) coming from activated macrophages (such as S100 proteins) may activate both the NFKB and the NLRP3 inflammasome pathways that drives the secretion of IL-1β and IL-18. In both diseases, additional triggering event such as disease flare, infection, in the context of genetic modifiers, may induce an amplification cascade of inflammation resulting in MAS. *Adaptive immune system*: In KD, the adaptive immune system may be activated primarily through antigen recognition by the BCR and TCR, and genetic variation in ITPKC may impair the regulation of calcium fluxes and then remove the inhibition of NFAT, which induces the secretion of IL-1α, a potent inducer of endothelial cells activation. In both Still’s disease and KD, NFKB activation induces an increased secretion of IL-6 and IL-17A, which participates in one case in bone erosion and inflammation and in the other case in endothelial inflammation and damage. Involvement of IgA from increased intestinal permeability and/or respiratory tract may participate to endothelial injury in KD. PAMPS: pathogen associated molecular patterns; DAMPS; danger associated molecular pattern; TLR; Toll-like receptor; NFKB; nuclear factor; NLRP3: NOD-like receptor family, pyrin domain containing 3; MAS: macrophage activation syndrome; BCR: B cells receptor; TCR: T cells receptor; ITPKC: inositol-trisphosphate 3-kinase; NFAT: Nuclear factor of activated T-cells; ROS: reactive oxygen species; NOD2: nucleotide-binding oligomerization domain 2.

**Figure 2 jcm-10-03244-f002:**
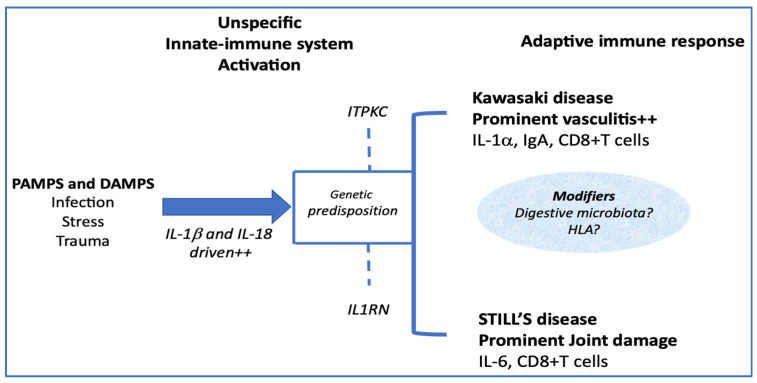
Modifiers that tip the immune response towards one clinical form or another.

**Table 1 jcm-10-03244-t001:** Clinical characteristics of Kawasaki disease compared with systemic JIA.

Clinical Sign	Still’s Disease	Kawasaki Disease
Estimated incidence	Children 0.6/<16 years [22] All ethnicities, minor prevalence and incidence variations	Japan: 309–330 cases per 100,000 <5 years [18] Canada (Ontario): 22 cases per 100,000 <5 years [19] Northern Europe: 5.4 to 11.4 cases per 100,000 <5 years [20]
Age of onset (Peak)	1–5 years [22]	2–5 years (1–2 years)
Male to female ratio	1/1	1.5/1
Family clustering	<1%	1%
Spiking fever	Yes almost 100%	Yes almost 100%
Skin rash	Diffuse macular, urticarial (75–80%)	Diffuse Macular, urticarial, scarlatiniform (80%) [23] Oedema and redness of the palms and soles (70%) [23]
Mucous lesions	Odynophagia (adults 66%)	Conjunctiva redness (89%) [23] Diffuse redness of the oral cavity (72%) [23] Strawberry tongue (56%) [23] Dryness of the lips (85%) [23]
Adenitis	Diffuse adenitis (42%)	Cervical adenitis (55%) [23]
Gastro-intestinal symptoms	Abdominal pain	Diarrhoea (60%) [23]
Hepato-splenomegaly	Children 20%	Hepatomegaly (56%) [23]
Arthritis	Persistent (25%)	Transient, and unusual
Heart involvement	Pericarditis (18%) Myocarditis Coronary aneurysms	Pericarditis (18%) [23] Myocarditis (3%) [23] Coronary aneurysms 25% of untreated patients
Macrophage activation syndrome	Apparent 10% Subclinical 40%	
Disease course	Monophasic with variable duration (40%) Polycyclic (10%) Persistent (50%)	Monophasic > 97% Recurrent (3%)
Response to treatment (yes/No)	[24]	
Corticoids	Yes	Yes
IV Ig	No	Yes
Anti TNF	Few	Yes
Anti IL-1	Yes	Probably yes, still exploratory
Anti IL-6	Yes	Unknown

**Table 2 jcm-10-03244-t002:** Biomarkers associated with disease activity in Still’s disease and Kawasaki disease and those associated with coronary aneurysms in Kawasaki disease [26,27,29,30,31].

Biomarker	Still’s Disease	Kawasaki Disease
CRP	Unspecific elevation	Unspecific elevation Very high levels are related to cardiac damage and IVIG resistance
Complete blood cell count	Polynucleosis, anaemia, and thrombocytosis are indicative of active disease Cytopenia are related to MAS	Polynucleosis, anaemia, and thrombocytosis are indicative of active disease Cytopenia are related to MAS
↑ d-dimer, ↓ fibrinogen	Indicative of MAS	Indicative of MAS
LDH, AST, ALT	Associated with disease activity	Associated with disease activity
Ferritin	Elevated, useful for diagnosis	Variably elevated
IL-6	Elevated, may be related with arthritis feature	Very high levels are related to IVIG resistance and coronary aneurysms
IL-1β	Associated with systemic symptoms	Elevated and related to cardiac damage
IL-18	Elevated in active systemic JIA and much higher in MAS	Elevated together with IL1β in active disease
TNFα	Normal or variably elevated	Elevated in acute phase
INFγ	Elevated in active systemic JIA and MAS	Very high levels are related to IVIG resistance and coronary aneurysms
S100 A12	Elevated in active systemic JIA and MAS	Elevated and corelating with disease activity
sRAGE *	Decreased during active disease	Decreased during active disease

* sRAGE: soluble receptor for advanced glycation end products.

## Data Availability

Not applicable.

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
