# Peer review of "Still’s Disease in the Constellation of Hyperinflammatory Syndromes: A Link with Kawasaki Disease?"

_jcm, 2021, doi:10.3390/jcm10153244_

Round 1
Reviewer 1 Report
I read with high interest the manuscript "Still's disease in the constellation of hyperinflammatory syndromes. A link with Kawasaki disease? submitted by Drusser and Kone Paut to the Clilical journal of Medicine.
I would commend the authors for this excellent work which sheds a relevant and well-argued light on this spectrum of inflammatory diseases in children. The description of the physiopathogical phenomena is clear and makes the reading of this manuscript accessible to a large public including clinicians interested in the subject.
I understand the similarities between Still disease and KD, however, and as the authors point out, endothelial involvement is arguably more prominent in KD, and the underlying mechanisms may be different, such as the involvement of angiopoeitins 1 and 2 in the phenomenon of endothelial activation and capillary leakage (Breunis WB Arthritis Rheum 2012; 64:306–315). Moreover, hemodynamic failures are, to my knowledge, more observed in KD than in Still's disease, both distributive shock and myocardial dysfunction (presumed interstitial intra myocardial edema). These severe failures at the acute phase are indicative of severe endothelial dysfunction and intense hyperinflammation. The authors could develop these aspects which emphasize the differences rather than the similarities between the two diseases.
Author Response
Dear Reviewer, thank you for your interest in our manuscript, and your overall good appreciation of its content. We have done our best to answer your queries, and also to respect the spirit of a rather short expert commentary, as expected by the editor.

Reviewer 2 Report
The reviewed manuscript has a very promising title. However, the principal objective of the study is not fully clear. Review article should broadly and carefully sum up the most current knowledge on the topic - the manuscript containing only 20 references with only 3 published in 2020 and 2021 does not fulfil that substantial criterion. A basic query for "Still's disease" in Pubmed provides 158 articles from 2020!
Abstract is hardly informative - it should attract potential readers with the hallmarks from the content of the manuscript.
Subsections 1 and 2 are completely incompatible with each other - the description of Still's disease is focused on the history and pathogenesis, whereas the characterization of KD is mainly regarding to the clinical picture. Furthemore, self-citation [14] is not applicable to the context.
Subsection 4 suddenly refers to COVID19, but there is no word about PIMS (MIS-C) in the text. I recommend the authors decide what they really want to describe here. The rest of the "Innate immune system activation" subsection is the strongest part of the manuscript, but one fine figure is not enough to cover the rest.
Figure 2 would be more informative as a comparison of similarities and differences (Subsection 3 vs. 5).
"Conclusion" should contain more clear take-home message what the authors have found and what they suggest for the future research.
Last but not least - English editing requires a thorough review, many sentences need to be rephrased to get the proper English word order.
Author Response
Dear reviewer, thank you for your time reading our manuscript. We understood that it did not meet your expectations, and we have done our maximum to improve it while respecting the requested format, which is not an extensive review but rather a short expert opinion.
Our answers points by points are in the added file.
With our best regards.

Reviewer 3 Report
interesting work on Stills disease and Kawasaki.
I miss a table on clinical signs and symptoms of both diseases: estimated incidence or prevalence, male-female gender ratio, familiary clustering?, age of initial signs and symptoms, arthritis yes/no, type of skin abnormalities etc, response to prednisone, IL-1 blockers, IL-6 bockers.
And I miss a table of immunology: role of inflammasome, IL-1, Il-6, TNF, risk of developing MAS etc
Author Response
Dear reviewer, Thank you for reviewing and suggesting improvement to our manuscript. As you suggested, we have added two tables describing the respective clinical and biological features of the two diseases.
Hoping theses changes will be at your convenience.
Best regards

Round 2
Reviewer 2 Report
I highly aprreciate the effort made by the authors to improve the quality of the paper and answer my previous doubts. In my view, Tables 1 and 2 are one of the strongest parts of the revised version and should definitely be moved from supplementary materials to the main content of the manuscript.
I still feel that a few sentences have inappropriate syntax but they do not dramatically decrease the research value of the presented paper. There are just a few last details to be altered before publication:
1) Keywords: the list is to broad and unordered. I recommend cutting it down to Kawasaki disease, Still's disease, disease overlap, interleukin-1
2) AA amyloidosis cases in children are very rare, in Still's disease I would even call them as casuistic. Therefore I think that there is too strong message in page 2, lines 53 and 60.
3) "Conceptual evolution of Kawasaki disease" - I really like a new paragraph about KD (page 2, lines 68-77), but the appropriate references are lacking there.
4) Page 3, line 130: "subclinically", not "sub clinically"
5) Page 3, lines 137: I am still not convinced by the authors why to mention COVID19 here. If the authors' objective is to stress that hyperinflammation may be caused by an infectious trigger, which may involve SARS-CoV-2 infection, I recommend rephrasing the first two sentences of the paragraph. In its present form, it only suggests that pandemic situation shed a new light to knowledge about aberrant innate immune response.
6) Page 6, line 222: Do the authors mean "coronary artery aneurysm" by CAA? The abbreviation is not defined in the text and the reference [29] involves a broader term "coronary artery lesions" abrreviated to CAL. I suggest rephrasing.
Author Response
Thank you for your comments.
We have added the table 1 and 2 in the main text as suggested.
Regarding English editing, the manuscript has been submitted to MDPI Author Services but will not be received before July 20.
1/ Keywords: we have corrected them as suggested
2/ AA amyloidosis: the sentence has been worked on: « these diseases having also in common, the young age of onset and the possible, although rare, progression to MAS and AA amyloidosis.”
3/ new paragraph about KD: references have been added as asked: “In 1961 in Japan, Dr Tomisaku Kawasaki, saw for the first time in a 4-year-old boy with a 2-week fever, non-purulent conjunctivitis, red, dry, and fissured lips, a raspberry tongue, cervical lymphadenopathy, erythema multiforme, and changes in the extremities (redness, swelling of the palms and soles). It was not until another 50 cases were described that a new disease was recognized and named after him [11]. The associated coronary artery aneurysms (CAA) were not identified until 1970. Initially, Kawasaki disease (KD) appeared to be an infectious disease, but to date, no single causative agent has been identified. Before intravenous immunoglobulins (IVIG) treatment became the gold standard, the incidence of CAA was 25% [12].
- Kawasaki, T. [Acute febrile mucocutaneous syndrome with lymphoid involvement with specific desquamation of the fingers and toes in children]. Arerugi Allergy 1967, 16, 178–222.
- Kato, H.; Koike, S.; Yokoyama, T. Kawasaki Disease: Effect of Treatment on Coronary Artery Involvement. Pediatrics 1979, 63, 175–179.
4/ Page 3, line 130: "subclinically", not "sub clinically": has been corrected
5/ We rephrased the sentence regarding COVID19 as proposed: “Both diseases involve a cytokine storm, a type of aberrant innate immune response as described in systemic forms of COVID19 (multisystem inflammatory syndrome) and shared by other as yet poorly understood conditions such as SAM, HLH, cytokine release syndrome, toxic shock syndrome and acute respiratory distress syndrome”
6/ coronary artery aneurysm = CAA : the term has been corrected as pointed out in the manuscript